



**On surface fluxes at night – the virtual chamber approach.**
Bruce B. Hicks[1], Nebila Lichiheb[2], Deb L. O'Dell[3], Joel Oetting[3], Neal S. Eash[3], Mark Heuer[2,4],
Latoya Myles[2]
[1]*MetCorps, PO Box 1510, Norris, TN 37828, USA*
[2]*National Oceanic and Atmospheric Administration, Atmospheric Turbulence and Diffusion Division, Oak*
*Ridge, TN 37831-2456, USA*
[3]*Institute of Agriculture, University of Tennessee, 2506 E.J. Chapman Drive, Knoxville, TN 37996, USA*
[4]*Oak Ridge Associated Universities, Oak Ridge, TN 37830, USA*
**Abstract**
Quantification of the emission rates of various gases from soils at night remains a challenge,
confronting climate science (in the case of $CO_2$ and $CH_4$) and agriculture science (for $NH_3$ and
$N_2O$, among others).  In sufficiently stable conditions at night, concentrations of such emitted
gases build up at the surface, with intermittent interruptions commonly attributed to the
passage of packets of turbulence.  The utility of conventional micrometeorological experimental
methods in such circumstances is questionable, and chamber methods have been developed to
meet the challenge.  Here, a statistical approach is proposed, in which micrometeorological
field data are used to replicate the likely characteristics of a chamber experiment, yielding
estimates of surface fluxes at the surface itself rather than at some height above it.  The
methodology proposed is developmental at this time, with details intended to correspond to
the use of both closed and vented chambers.  Its application to three recent field studies is
explored:  (1) a study of nocturnal $CO_2$ emission from two test areas (one previously tilled and
the other not) in Ohio in 2015; (2) a similar experiment conducted in Zimbabwe in 2013 (one
area previously tilled and a second left fallow), and (3) an investigation of $NH_3$ effluxes from a
crop previously treated with urea ammonium nitrate (UAN), in Illinois in 2014. There are few
measurements with which to compare the results presented here, however the values obtained
are within the range of available field data.
**Keywords:**  Soil efflux, $CO_2$, $CH_4$, $NH_3$, nocturnal intermittency, Zimbabwe





## 1. Introduction

Quantification of the emission rates of trace gases from soils in fields, wetlands and forests presents a problem that standard micrometeorological methods fail to solve (Skiba et al., 2009; Wilson et al., 2012). While eddy correlation techniques, in their various forms, have gained popularity, their requirement for sufficient fetch remains an obstacle that is difficult to overcome, especially at night (Aubinet, 2008). Bowen ratio methods are less susceptible to fetch limitations, because relevant measurements can be made at a lower height than for eddy correlation or flux/gradient calculations (Meyers et al., 1996).

Measurement of fluxes at night is especially demanding (Schneider et al., 2009; Darenova et al., 2014). While sensitivity to fetch limitations is reduced, the Bowen ratio methods remain fallible at night, when the inherent assumption that gradients and fluxes are closely associated is vulnerable. To address the matter of emissions from soils at night, field programs often rely on measurement methods of an entirely different kind – the use of chambers that confine emissions from the ground within closely-monitored volumes and thus eliminating the problems associated with fetch. In the case of carbon dioxide ($CO_2$), for example, the rate of increase in $CO_2$ concentrations within such a confined volume is an indication of the flux from the surface. However, it is recognized that the presence of any such chamber imposes an obstacle to the natural flow regime, with consequences that are hard to quantify. On the whole, there is no method that appears to satisfy the objections of all critics.

Comparisons among results obtained using chambers of different configurations have been revealing. Comparison of results from closed ('static', q.v. Edwards, 1982) chambers and alternative 'dynamic' approaches (Norman et al., 1992) has received particular attention. Field



studies summarized by Nay et al. (1994) have indicated considerable differences, sufficient that
laboratory tests were conducted of their performance involving the measurement of $CO_2$ efflux
rates of known magnitude from test surfaces. The laboratory evaluations confirmed the level of
uncertainty derived from the many field comparisons, with differences sometimes exceeding a
factor of two. A more extensive examination was reported by Pumpanen et al. (2004), whose
independent research allowed them to conclude that "Any use of the static-chamber method
ought to be particularly scrutinized."  Wang et al. (2004) compared results from closed and
vented chambers, with results indicating differences in derived soil efflux rates (of ammonia
($NH_3$) in their tests) ranging up to a factor of five.
Here, a statistical approach is proposed, replicating the constraints associated with chamber
methods in a way that leads to estimates of average fluxes rather than specific short-term
situations. The present intent is to demonstrate the utility of the methodology, without
proposing that it should replace other experimental methods but indicating the benefits of a
statistical way of looking at the results of field studies.
The concept of a 'virtual chamber' analysis to derive flux information from nighttime
concentration data (of trace gases like $CO_2$, methane ($CH_4$), nitrogen oxide ($NO_x$), and $NH_3$)
arose in examination of data obtained in Zimbabwe, which illustrated the ramp structure at
night considered due to dilution of $CO_2$ accumulation in the stratified layers of air near the
surface by nocturnal intermittent turbulence (Hicks et al., 2015).  The Zimbabwe dataset is of
special interest, because it relates to a field site on the arid Zimbabwe plateau near Harare, at
an altitude of more than 1400 m asl.  The Zimbabwe dataset will be revisited below, as one of
the three examples of the analysis methodology now proposed.



## 2. The virtual chamber

Consider the case of trace gas emission from a specific surface. At this point, there is no consideration of the conventional requirement for time stationarity and spatial uniformity. These issues will be considered later. In daytime convective (and unstable) situations, the measurement of the fluxes is typically considered as a standard micrometeorological exercise. At night, however, the constraints imposed by the necessary assumption that fluxes measured in the air at some convenient height above a surface of special interest are indeed representative of that particular surface presents substantial obstacles (Aubinet, 2008). It is the nocturnal stratified atmosphere in contact with the surface that will be considered here.

There are many published examples of nighttime time series of measurements of concentration of some gas (e.g. $CO_2$ or $CH_4$) in ground-level air that displays a saw-tooth pattern, with slowly increasing concentrations interrupted by sharp decreases (Wehr et al., 2013). These intermittent decreases are commonly attributed to bursts of turbulence interrupting an otherwise quiescent surface boundary layer. There are several possible causes of these turbulent events, such as the oscillations of a low-level jet or the generation of gravity waves by some upwind obstacle (Aubinet, 2008; Mahrt et al., 2019). It is possible that the phenomena are a basic feature of strongly stratified flow (Manneville and Pomeau, 1979; He and Basu, 2015), as a consequence of interactions among different processes (Lorenz, 1963). The related phenomena are almost invariably external to the classical micrometeorological framework, in which turbulence is associated with the characteristics of the local surface. The optimal time resolution is therefore not associated with conventional micrometeorological examinations of fluxes and gradients, but instead short enough to identify with clarity occasions of



intermittency so that these can be excluded from the analysis now proposed -- intended to
focus on the causes of increases in surface concentrations. Field experience indicates that a
time resolution of the trace gas concentration record should best be such that events shorter
than five minutes can be resolved.
Suppose a fast-response anemometer is deployed at some convenient height, providing three-
dimensional velocity data (means and variances) every five minutes, or over some alternative
averaging time deemed appropriate. Simultaneously, measurements of concentration (C) of
some atmospheric trace constituent are made, at some point below the sonic installation. After
data accumulation extending over many days, consider the statistical characteristics of
ensembles of data generated after sorting according to time of day. A partial correlation
examination of three variables is of present relevance:
$$X_1 = dC/dt \qquad\qquad (1)$$
$$X_2 = u \qquad\qquad (2)$$
$$X_3 = \sigma_w \qquad\qquad (3)$$
where notation is conventional.  In practice, the wind speed u is an output of the sonic
anemometer, as is the standard deviation of the vertical wind component $\sigma_w$.  The rate of
change of concentration, dC/dt, is conveniently computed from the initial time sequence of
measurement as:
$$dC/dt = (C_{n+1} - C_{n-1})/(t_{n+1} - t_{n-1}) \qquad\qquad (4)$$





A first-order partial correlation analysis (or multiple regressions) yields the best-fitting
coefficients in a relationship of the kind:
$$X_1 = a_x + b_{x12} \cdot X_2 + b_{x13} \cdot X_3 \qquad (5)$$
The intercept $a_x$ is therefore the value of $X_1$ (i.e. dC/dt) that would be expected in the case for
which $X_2$ and $X_3$ were both zero; i.e., for when there is no effect of the wind (no advection) and
no turbulent exchange in the vertical at the level of the anemometer ($z_a$). The situation then
envisioned is that of a conventional closed-chamber experiment, but lacking the consequences
of a physical presence that could influence the natural circumstances.
Figure 1 presents a schematic illustration of the construct now considered. Two configurations
are illustrated. Consider, first, the closed-chamber option as discussed above and as illustrated
to the left of the diagram. Clearly, the assumption that a positive value of dC/dt represents the
accumulation of trace gas in the stable layer of relevance warrants examination. In concept,
the quantity C would best represent the average within the virtual chamber so defined, of
cross-sectional area of 1 m$^2$ and of height $z_a$. While this conceptual entity is defined in terms of
specific measurable dimensions, its relevant characteristics are now based on the statistical
extrapolation of other observations.



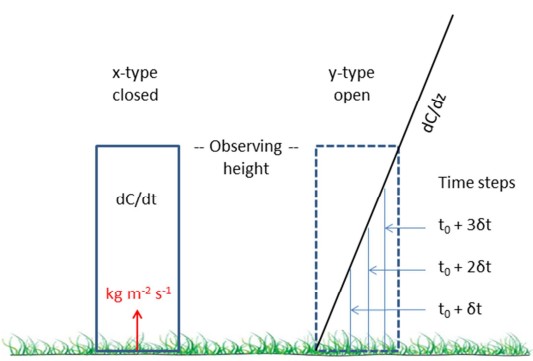


Figure 1. A schematic illustration of the concepts proposed. The x-type chamber

simulation is represented to the left, leading to an approximation that the efflux at the

surface can be derived from measurements of the rate of change of concentration with

time. This is considered an extreme case for the limiting statistical analysis now

proposed. An alternative y-type extreme is represented to the right, in which the depth

of the layer of relevance is allowed to grow while maintaining the same concentration

gradient.


The assumption that dC/dt is the appropriate revealing quantity in the present closed-chamber
context requires further attention. It is considered here to represent an extreme circumstance
controlling the statistics that follow. An alternative extreme might well simulate an open
chamber, in which the depth of the affected layer increases with time, according to the flux
from the surface, but maintaining a constant gradient in the air.  In this alternative hypothetical
case, the concentration observed at some height at or below $z_a$ will increase as the square root
of elapsed time (as is illustrated in Figure 1), rather than linearly as required by the closed-



chamber approach.  Hence, the conceptual virtual chamber can be considered in two ways,
representing extremes.  The first ('x-type') makes use of dC/dt as a key variable, with
conceptual association with the operation of a closed but stirred chamber.  The second ('y-
type') is intended to simulate the characteristics of an open chamber, by substituting $Y_1$ =
$(dC/dt)^2$ for $X_1$ = dC/dt in the discussion above (specifically in Equation (1)).  In this second case,
the eventual relationship sought is
$$Y_1 = a_y + b_{y12} \cdot X_2 + b_{y13} \cdot X_3 \qquad (6)$$
which replicates Equation (5).  The two separate estimates of the ensemble-mean average
fluxes are then
$$F_x = z_a \cdot a_x. \qquad (7)$$
and    $$F_y = z_a . a_y^{0.5}/2 \qquad (8)$$
where the divisor arises from the consideration of a right-triangular conceptual configuration in
the second case (as is evident in Figure 1), rather than the rectangular figure that contains it in
the former.
Regardless of the assumption adopted, a measure of the depth of the layer of relevance is a
central requirement.  Here, this depth is assumed to be the level at which the loss vertically is
indicated to trend to zero – the height of measurement of $\sigma_w$ or of some other convenient
indicator of minimal vertical turbulent exchange.  Virtual temperature gradients or TKE could be
used equivalently.  (Note, however, that the intent to consider the limit as transfer to air above
trends to zero requires that the temperature gradient variable of relevance should be the





inverse of the virtual potential temperature gradient, i.e. determining the limit as $\delta Tv$ tends to
infinity.)  The inclusion of wind speed is in recognition of the desire to eliminate advection as a
major causative property, even though if the site in question is sufficiently uniform the wind
speed contribution would be expected to become negligible.  If fetch is limited and if the flux of
interest can be associated solely with that fetch, then $u/X_f$ becomes an attractive variable,
where $X_f$ is the upwind fetch.
In the analysis to follow, two distinct methodologies are proposed.  The x-analysis as outlined
above assumes that changes in concentration in air near the surface can be considered as being
proportional to changes in the surface fluxes.  The corresponding y-analysis replaces $X_1$ by $Y_1 =$
$X_1^2$ (q.v. Figure 1) and assumes that changes in concentration measured at some specific height
are determined by changes in the depth of the surface stable layer, such that the concentration
gradient remains the same.
In practice, the requisite analysis employs standard statistical methods, adapted from textbook
examples (in which matrix algebra is commonly employed) for the present simple case by
evaluating the correlation coefficients of relevance ($R_{12}$ is the correlation coefficient between
variables $X_1$ and $X_2$) and the resulting partial correlations ($R_{12.3}$ is the partial correlation between
variable $X_1$ and $X_2$ when the influence of variable $X_3$ is accounted for). Of considerable relevance
in analyses like that presented here are the consequent quantities
$$R_{1.23}{}^2 = R_{12.3}{}^2(1 - R_{13}{}^2) + R_{13}{}^2 \qquad (9)$$
$$R_{1.32}{}^2 = R_{31.2}{}^2(1 - R_{12}{}^2) + R_{12}{}^2 \qquad (10)$$



which quantify the proportion of the variance in variable $X_1$ that can be explained by the
combination of variables $X_2$ and $X_3$.   Finding the equality evident in the two ways of deriving
this quantity is a confidence-building exercise of some considerable satisfaction.
Standard statistical relationships lead immediately to the quantification of the variables $a_x$ and
$a_y$ in Eqs. (5) and (6).  Estimates of the effluxes then derive immediately, assuming that the
height of measurement determines the average height used to define the conceptual chamber
of relevance.  This is a statistical matter that invites further examination.
In neither the closed-chamber or the open-chamber approximation can the results be
considered actual measurements of the surface efflux rates. They are no more than statistical
estimates of these fluxes, based on numerically quantified heurism. Conventional experimental
campaigns typically provide bodies of suitable data. In the following, three examples of recent
application of the virtual chamber approach will be described. The first of these relates to a
study of the consequences of tilling on the emissions of $CO_2$ from an agricultural surface in
central Ohio (O'Dell et al., 2018). The measurement program was based on standard Bowen
Ratio Energy Balance protocols. The second repeats the analysis, using similar data derived
from an earlier study conducted in Zimbabwe (Hicks et al, 2015, O'Dell et al, 2015). The third is
based on a study of ammonia emissions from an area previously treated with urea-ammonium
nitrate (UAN) as a nitrogenous fertilizer in central Illinois (Nelson et al., 2017, 2019). These
examples were selected for present attention because the dimensions of the subject areas are
sufficiently small that conventional nighttime micrometeorological methods are not likely to be
productive.



### 3. The Ohio study of tilling

Figure 2 (a) shows the layout of the field site of the Ohio study (O'Dell et al., 2015).
Measurements were made of the concentration of $CO_2$ in the air, the air temperature and
vapor pressure at two levels close to the surface, wind speed, and the surface temperature as
reported by downward looking infrared thermometers. The central question related to whether
previous tilling of the surface exacerbated $CO_2$ emissions from the soil at night. The
experimental program involved the collection of data from two adjacent areas (as seen in
Figure 2), each about 200 m x 200m in size. One of these test areas was tilled (#1), and the
other not tilled (#2) before maize (*Zea mays* L.) was planted. The data were collected in
November 2015, during the immediate post-harvest period. Observations used here were
obtained centrally within each of the study areas, with a time resolution of five minutes.
Experimental constraints caused this resolution to be adjusted, so that the data used here
represent ten-minute intervals.

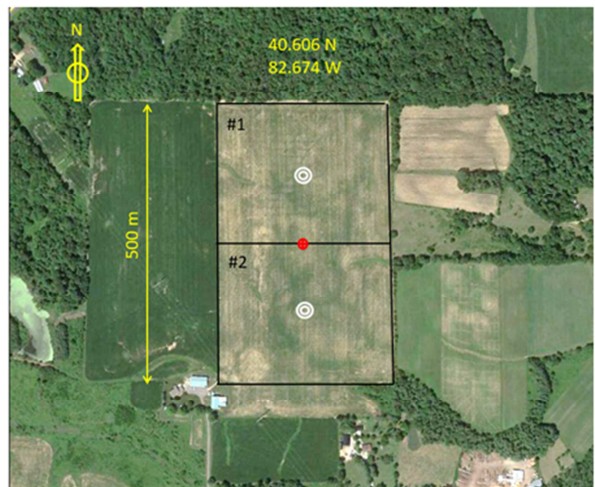





*Figure 2. The surface layout of the test areas of the Ohio (2015) field study. Stars indicate*
*the locations of measurement systems.  The image is derived from copyright © Google*
*Earth.*

The selection of data for use in the present study has been based on the requirements of the
concepts involved. First, nighttime data must be considered. Hence, data records with reported
positive net radiation have been excluded. Further, the intent was to interpret the increases in
concentration observed near the surface in stable conditions. To this end, situations in which
$dC/dt < 0$ have been rejected (since such situations are likely the consequence of nocturnal
turbulence intermittency, another controlling mechanism to be considered elsewhere). In the
absence of sonic anemometer data, measurements of the virtual temperature gradient derived
from the conventional Bowen ratio energy balance (BREB) methodology have been used.  To
quantify the limit as vertical mixing approaches zero, $X_2$ has been taken to be $(\delta\theta_v)^{-1}$, where $\delta\theta_v$
is the difference between levels $z_1$ and $z_2$ (above the ground) of the measured virtual potential
temperature.  In the present case, $z_1$ is about 0.3 m and $z_2$ about 1.8 m, so that the effective
height to be used in the estimation of fluxes from the evaluations of $a_x$ and $a_y$ is about 1 m.

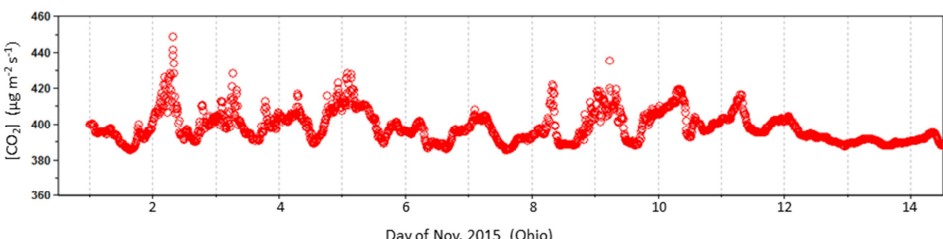




*Figure 3.  Time trends of $CO_2$ concentration at a height of about 1 m above a recently*
*harvested maize field in Ohio.*
Figure 3 presents a sample time record of $CO_2$ concentrations from the Ohio experiment,
obtained above the untilled field. The characteristic nighttime concentration build-ups are
obvious, as are the consequences of intermittent turbulence. The mid-American farmlands
(within which the present Ohio field site was located) are notorious homes of frequent
nocturnal jets, with consequences that include the generation of irregular bursts of turbulence
(q.v. Blackadar, 1957; Banta, 2008). It is assumed here that it is such irregular bursts of
turbulence that curtail the otherwise steady accumulation of $CO_2$ in the atmospheric boundary
layer adjacent to the surface. The present intent is not to investigate these bursts of
turbulence, but instead to accept them as features of the nighttime atmospheric environment
and then to examine the trends with time when they are not dominant factors.
Figure 4 summarizes results obtained from application of the virtual chamber analysis
methodology outlined above.  In Figure 4 (a), plots are shown of the proportion of the variance
in $X_1$ and likewise in $Y_1$ that can be accounted for by consideration of variables $X_2$ and $X_3$
(following Eqs. (9) and (10)). If this proportion is close to unity, then the data constitute a sound
basis for examination in the way now suggested. However, such high values are not often
encountered in the surface boundary layer atmosphere. For example, the relationship between
wind speed and the surface stress is usually quantified by a correlation coefficient of the order
of 0.4, so that less than 20% of the variance in stress is accounted for by changes in the wind
speed.  In this light, the values plotted in Figure 4 (a) are somewhat reassuring, ranging from



about 20% to 40%. It is noticeable, however, that the values associated with the open chamber
(y-type) assumption are lower than those of the closed chamber kind (x-type).
Figure 4 (b) presents the estimates of surface effluxes derived from the present analysis. There
being no obvious reason to prefer one of the two kinds of analysis rather than the other, it is
presently preferred to accept both and to view them as extremes. It could be argued that
Figure 4 (a) indicates that x-type must be preferred to y-type, but a conclusion of this kind
would require multiple tests and is certainly premature at present. The best estimate of the
average surface emission rate is therefore likely to be the average of all of the values plotted:
For the tilled surface, $1.70 \pm 0.31$ $\mu$g m$^{-2}$ s$^{-1}$, and for the untilled, $1.33 \pm 0.08$ $\mu$g m$^{-2}$ s$^{-1}$. The
most likely averages of $CO_2$ nocturnal emissions from Ohio agricultural soils are therefore
indicated to be about 2 $\mu$g m$^{-2}$ s$^{-1}$ for the conditions of the current test, regardless of whether
the surface was previously tilled.

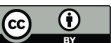

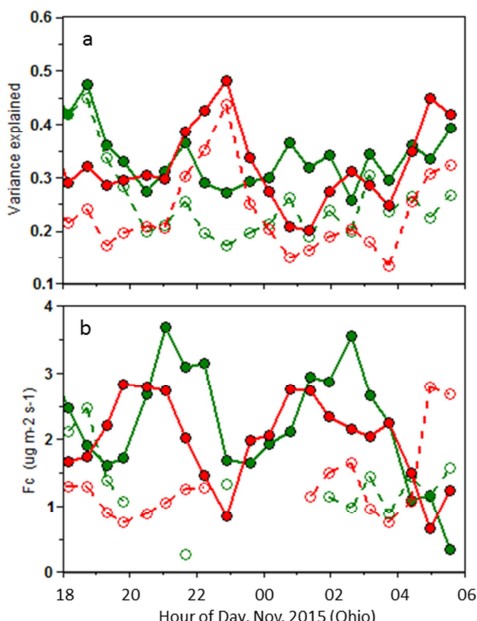


*Figure 4. (a) The variation with time of the proportion of variance in dC/dt (= X1; 'closed*
*chamber') and (dC/dt)^2 ( = Y1; 'open chamber') that can be accounted for by statistical*
*consideration of the virtual temperature gradient (employed as its inverse, 1/δTv) and*
*wind speed (u), for the two areas of present interest – one tilled before seeding (green)*
*and the other not tilled (red). (b) the estimates of surface effluxes derived from the same*
*analysis. As elsewhere, solid points indicate results obtained using the closed-chamber*
*approximation described here, open points represent vented chamber approximations.*

**4. The Zimbabwe plateau data**




A field examination of the comparative benefits of various agricultural practices was conducted
in Zimbabwe, starting in 2013 (O'Dell et al., 2015, Hicks et al., 2015). Identical BREB
instrumentation was set up centrally in four neighboring experimental areas, of which two will
be considered here (identified as #1 and #2 in the site depiction in Figure 5).

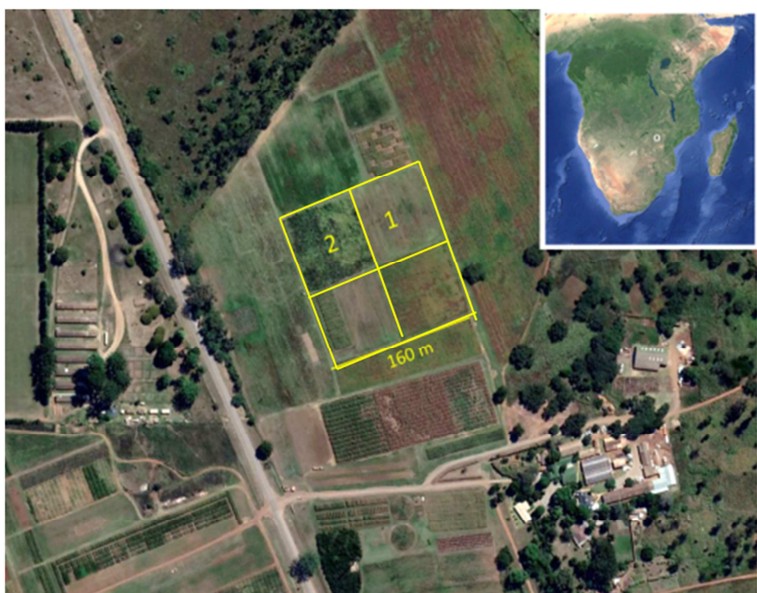


*Figure 5. The field site of the Zimbabwe study, showing the two test areas, 1 and 2,*
*considered in the present analysis. At the time of the analysis to follow, area 1 was*
*fallow and area 2 was plated with maize. The inset locates to field site within the*
*African continent (31.021E, 17.722S). Both images derive from copyright © Google*
*Earth.*



An earlier analysis focused on the nocturnal data obtained, with a time resolution of five
minutes so that occurrences of intermittent turbulence were readily apparent (Hicks et al.,
2015). In the lack of major upwind surface irregularities, these occurrences were attributed to
the gravity wave phenomenon considered in detail by Blackadar (1957) but Mahrt et al. (2019)
show that there are several alternative causative mechanisms. The key point of the Zimbabwe
finding was that the site in question was at an altitude of more than 1400 m asl, and the
occurrence of the turbulence intermittency phenomenon at this altitude is a revealing
indication of the ubiquity of the mechanism. The data collection protocols used in the
Zimbabwe study were the same as were used in the Ohio experiment, discussed above.  The
analysis now considered for the Zimbabwe dataset mirrors that discussed above for Ohio.

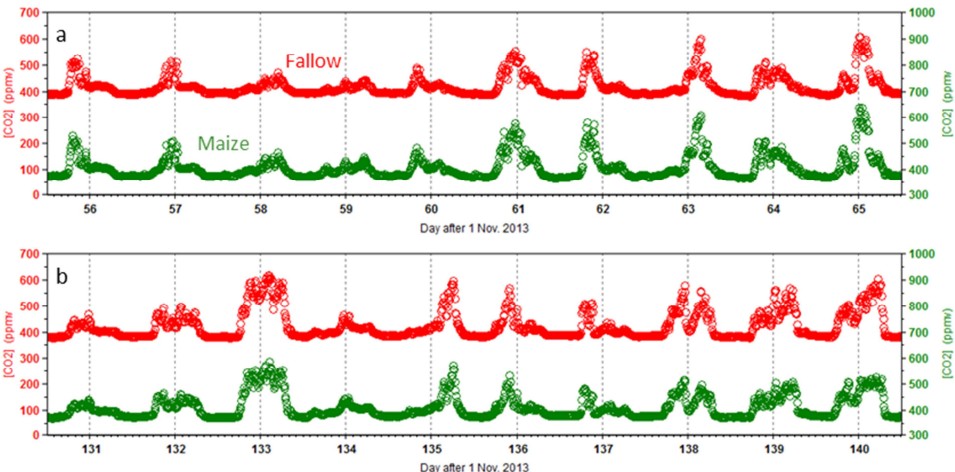


*Figure 6.  Data from the Zimbabwe field site, illustrating the repeated occurrence of a*
*nighttime saw-tooth pattern for two widely-separated periods (selected at random from*
*the overall six-month data record).  Values plotted represent ten-minute average $CO_2$*



*concentrations.  Red represents data for the fallow field (#1 in Figure 1), and green the*
*adjacent field (#2) carrying a growing crop of maize.  No data have been omitted.*

Figure 6 presents two sequences of $CO_2$ concentration measurements, obtained above the two
fields of current interest at an average height of about 1 m above the vegetation or soil.  The
periods selected for presentation here were selected at random, but are intended to show the
similarity in the overall behavior of the growing maize and the fallow field.  Note, however, that
the fallow field carried a coverage of flourishing native weeds, so that any difference could well
have been obscured.
Figure 7 replicates Figure 4 using the Zimbabwe data. In Figure 7 (a) it is seen that the
proportion of variance explained is generally low, except for a peak centered on midnight. The
interpretation of this is that the efflux conclusions based on the two-hour window around
midnight are the most robust.  In Figure 7 (b) it is clear that the flux estimates are well behaved
for this period, with an average of about 20 $\mu g\ m^{-2}\ s^{-1}$ of $CO_2$ emission.  As before, there is no
convincing reason to prefer the y-type results over the x-type, even though the negative results
(x-type) indicated in the diagram are disturbing. If all of the results are averaged (as was the
case in consideration of the Ohio dataset), the resulting estimate of the $CO_2$ efflux for the
Zimbabwe November data is 11.1 ± 1.3 $\mu g\ m^{-2}\ s^{-1}$ for the area sown with maize, and 10.3 ± 1.5
$\mu g\ m^{-2}\ s^{-1}$ for the fallow. At the time of these measurements, the maize had not yet fully
emerged and the fallow field was poorly vegetated (with weeds). Concentration and virtual
temperature data refer (as before) to a height of about 1 m above ground level.



The test areas used in the Zimbabwe study were smaller than those of Ohio:  80 m on side in
comparison to 200 m. It must be expected that this difference in size will have an effect on the
conclusions derived from the present analyses, although the procedure is designed to be based
on extrapolation of observations to a situation in which the wind speed is zero, at which point
fetch considerations become meaningless.

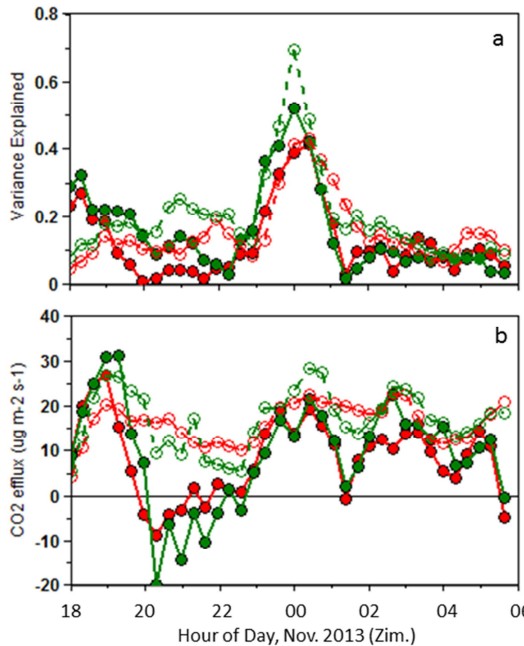


*Figure 7.  As in Figure 3, with red points relating to a fallow field and green to an*
*adjoining area recently planted with maize (on 8 November, 2013).  The dataset*
*development in this case differs, with fewer points making up each ensemble and with*
*consequent increased scatter in the results.  The period represented here is the entire*
*month of November. Solid points indicate results obtained using the closed-chamber*
*approximation described here, open points represent vented-chamber approximations.*

**5. The Illinois NH₃ study**
Nelson et al (2017) reported a study of NH₃ fluxes from an area previously treated with UAN as
a nitrogenous fertilizer. Such treatment is a common practice within the agricultural community
particularly in the Midwestern US. The site is illustrated in Figure 8.

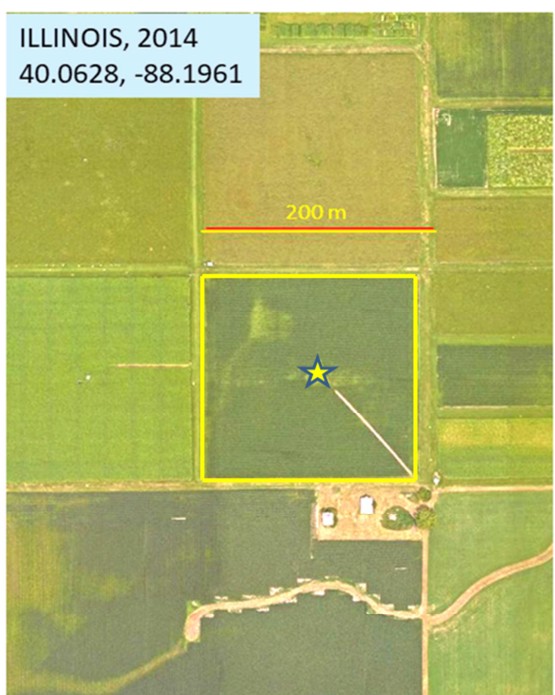




*Figure 8. As derived from copyright © Google Earth, a map of the 4-ha field site used in*
*the Illinois (2014) study of ammonia fluxes following fertilization using urea ammonium*
*nitrate.*

Classical investigations of this issue have relied on gross measurements of changes in soil
nitrogen content, typically over periods of weeks. The results of such measurements are highly
influenced by external factors, especially rainfall. The Illinois study of interest here was
intended to explore the micrometeorological use of new fast-response ammonia sensors. In the
process, time series of $NH_3$ concentrations near the ground were derived, illustrated in Figure 9,
which constitute a basis for exploration using the virtual chamber methods now proposed. Like
the Ohio experimental area considered above, the Illinois site is within the mid-American
farmlands and is subject to characteristic nocturnal jets and consequent bursts of turbulence
occurring at night (as have been investigated in detail by Banta, 2008, for example). Ammonia
gas measurements made at the Illinois site in 2014 reveal precisely the cyclical pattern
expected to result from such turbulence intermittency, as is seen in Figure 9.  The opportunity
exists, therefore, to make use of the analytical methods suggested here in order to derive
information regarding the rate of emanation of ammonia gas from the previously fertilized
area, so as to derive flux data not influenced by rainfall itself but such that the influence of
factors like soil moisture content and temperature could perhaps be assessed.



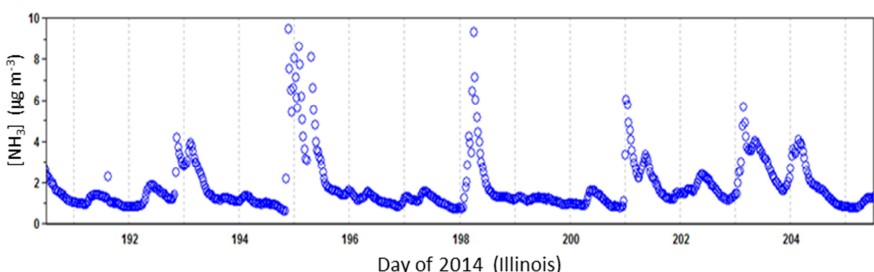


*Figure 9.  A time sequence of NH$_3$ concentration measurements obtained in the 2014*
*Illinois field study.*

Using a two-week period of data (of which Figure 9 is representative), results obtained using
the virtual chamber approach are as indicated in Figure 10.  A key distinction between the
present ammonia case and the CO$_2$ examples considered above is that the NH$_3$ concentrations
are available as 30-min averages, instead of the 10-min averages used for the Ohio and
Zimbabwe datasets.  The consequences of this are apparent in Figure 10 (a), where the total
proportion of variance explained in the rate of change of NH$_3$ concentration with time is lower
than that derived for the CO$_2$ cases.  It is clear that the methods considered here require finer
time resolution than is common for conventional micrometeorological studies, since the overall
intent is now to detect and omit situations in which intermittent bursts of turbulence affect the
buildup of concentrations in the layer of stratified air in immediate contact with the surface.
Reliance on data that fail to permit fine distinction between periods of turbulence bursting and
the quiescent periods between successive intermittent bursts occurrences certainly obscures
the statistical correlations on which the present techniques are based, and will result in an
underestimation of the efflux in question.

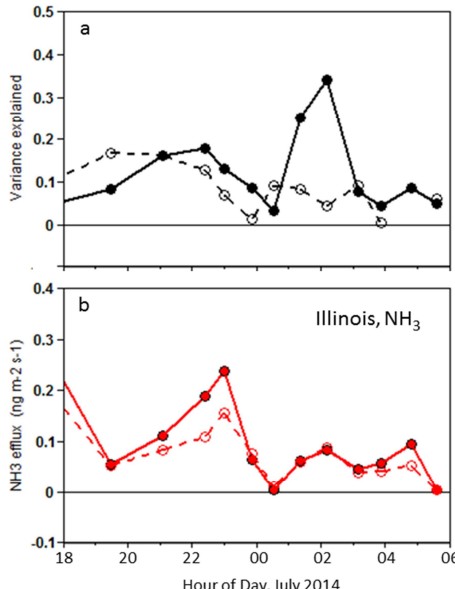


*Figure 10.  Results derived from measurements of NH₃ above a field previously fertilized*
*with treatment of UAN, in Illinois in 2014.  As in the similar presentations above, (a)*
*represents the total proportion of variance explained in dC/dt by the combined*
*influences of wind speed and turbulent mixing, from interpretation of which (b) indicates*
*how derived NH₃ effluxes vary through the night.  As before, solid points indicate results*
*obtained using the closed-chamber approximation described here, open points represent*
*vented chamber approximations.*

**6. Discussion**



The present intent has been to make use of several different datasets to illustrate the potential
utility of the virtual chamber analytical methodology and not to focus on results from any
specific location in detail. Nevertheless, it is clear from the analyses above that the $CO_2$ efflux
from the Zimbabwe site exceeded that from the Ohio location by about an order of magnitude.
The reason is not clear, but several obvious considerations are worthy of attention. For
example, differences in soil temperatures could explain the difference: about 7.5 C for the Ohio
dataset and 24.0 C for the Zimbabwe. Differences in soil moisture are to be expected, and
would likely contribute to the difference. The Ohio surface was covered with the detritus of
recent harvesting, whereas the Zimbabwe surface had been recently planted. All in all, the
situation is complicated and requires more attention than is presently appropriate.
Irrespective of the negative consequences imposed by the half-hour sampling associated with
the Illinois dataset, Figure 10 (b) provides a convincing indication that the rate of $NH_3$ emission
from the ground was about 0.1 ng $m^{-2}$ $s^{-1}$.  Wang et al. (2004) report results that indicate that
the rate of volatilization of ammonia from an area bearing a maize crop depended almost
linearly on the amount of urea previously broadcast. The maximum $NH_3$ emission rate was
about two days following application of the fertilizer, but at an average rate of from 0.1 to 0.8
kg $ha^{-1}$ $d^{-1}$, corresponding to about 0.1 to 0.8 μg $m^{-2}$ $s^{-1}$. The efflux estimate derived from the
present analysis is three orders of magnitude lower. In all comparisons of this kind, it should be
remembered that the classical chamber study results are typically presented as whole-day
averages, whereas the virtual chamber results bow being considered represent only those
times of the day when the air in contact with the surface is stratified – usually, at night. Once
again, the difference invites investigation.




## 7. Conclusions

The methodology presented here diverges substantially from familiar micrometeorological
strategies. First, it is focused on the ground itself (or the vegetation above it), and does not rely
on the assumption that measurements made above the ground are indicative of the local
surface. Second, the reliance on statistical methods to drive the analysis towards situations in
which the prevailing stability is high but the wind speed is zero reduces (if not eliminates) the
conventional requirement regarding large fetches. Third, the method requires measurements
with a sufficient time resolution (less than 5 min) such that the effects of intermittent bursts of
turbulence can be identified and eliminated. This is in direct contrast to standard
micrometeorological practice, which requires a sampling duration long enough that a
statistically significant sampling of these same bursts of turbulence can be obtained.
However, the methods now presented do not result in a defensibly deterministic quantification
of the relevant surface fluxes. It is assumed that the two alternative methods presented and
discussed above represent extremes, so that the exchange rates of actual relevance lie between
the corresponding bounds. A similar line of thinking was proposed by Wang et al. (2004), who
report on results obtained from field studies over the North China Plain using both closed and
vented chambers. These two experimental methods yielded flux estimates that differed by as
much as an order of magnitude. Hence, the differences found in the studies now considered
appear reasonable, although clearly requiring additional research.



**Acknowledgements**


The authors gratefully acknowledge funding from the National Science Foundation (Award
Numbers AGS12-36814 and AGS 12-33458) for the Illinois study. The scientific results and
conclusions, as well as any views or opinions expressed herein, are those of the authors and do
not necessarily reflect the views of NOAA, the Department of Commerce, or NSF.



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
