# Peer review of "On surface fluxes at night – the virtual chamber approach."

_Biogeosciences, 2019_

## Referee Comment (RC1) · Anonymous Referee #1 · 13 Dec 2019

The authors propose a novel approach to estimate fluxes exchanged between the surface and the atmosphere in nocturnal conditions. The method is based on a so-called Âń virtual chamber approach Âż, which consists basically to estimate the storage term every five minutes and to infer its value in the absence of advection and turbulence. Under such conditions, it is supposed to equal the flux.

As estimating fluxes during nocturnal conditions remains a big challenge, there is a clear need for new approaches. The one proposed here has the advantage to not require additional measurements as those already provided by current flux towers. Only a recalculation of tracer concentrations and wind velocity means and variances on a five minute basis is required. It could thus be implemented at many different sites on existing data sets. However, I see two big weaknesses in the approach, which let

me think that it is not yet mature and that a strong reappraisal of the method is needed.

First, if I understand well, the method relies on the hypothesis that vertical gas concentration profiles are flat (in the closed chamber option, Eq. 7), or linear (in the vented chamber option, Eq. 8). This is not realistic, vertical concentration gradients are generally much higher close to the surface than upward (see, for instance, profiles in Aubinet et al., 2005). This suggests that the method cannot be applied as it is in forests (where the problem of night flux estimate is the most critical) and is also probably questionable over shorter vegetation. Replacing a single point concentration by a profile measurement could certainly improve the method (Nicolini et al., 2018).

Secondly, the method is not validated. The authors present flux estimates based on three measurement campaigns but they recognize that their method clearly underestimates the fluxes in the third campaign, due to a too low data acquisition frequency, and that the values they propose in the two first campaigns are only orders of magnitude. I think that a better validation could be provided by comparing their estimates with eddy covariance flux estimates captured during turbulent nights at the same site.

In addition, I'm surprised by the numbers they propose: at the Ohio site, their approach provides a flux estimate of 1-2 $\mu$gm-2s-1 which is 15-30 times lower than the average respiration rates observed at Fluxnet sites (Baldocchi et al., 2018). This is not totally impossible but appears in contradiction with another publication by the same team: O'Dell et al (2015) indeed reported that, at the same site, one plot emitted 146 g $CO_2$ m-2 on 104 days, which would correspond to an average of 16 $\mu$gm-2s-1. At the Zimbabwe site, the authors obtained an average of about 20 $\mu$gm-2s-1 , which is a rather realistic order of magnitude ; however, here again, the same team (O'Dell et al, 2018) reported at this site emissions of 197 and 235 g m-2 over 139 days, which corresponds to a three times higher average. It is clear that these values could not be compared directly as they are not taken at the same time scale. However, in view of the differences in the orders of magnitude, no indication is given that the virtual chamber approach provides reasonable flux estimates but I'm rather pushed to think that

it underestimates the fluxes, which would be logical if vertical concentration gradients are not taken into account. A more detailed validation and, possibly, refinements of the method are thus needed, which seems possible in view of the available datasets from the Fluxnet or the ICOS networks.

I have also a remark concerning the statistical treatment: partial correlation coefficients are computed and their time course is presented in the results but there is no mention on how they are taken into account in the study. Are they used as quality criteria? If yes, which thresholds are considered? Moreover, how do the authors justify the choice of these specific criteria (in place, for example of the standard error of ax and ay coefficients, which would provide a direct estimate on the flux uncertainty)?

I have no specific comments on the paper structure and writing, which are both good. Just one remark concerning figure 1b: I suppose that the black line represents the CO2 concentration evolution with height and not its gradient as indicated on the line (the gradient is the slope of the line). This should be clarified.

―――――――――――――――――――――

---

## Referee Comment (RC2) · Thomas Foken (Referee) · 27 Mar 2020

The authors present an interesting approach of using a "virtual chamber" to discuss the influence of the complicated condition of the atmospheric turbulence at night on chamber measurements. This is an important contribution to making chamber measurements more representative, even if some problems are still open like the validation. Furthermore, the design of experiments for virtual chambers must be updated. The three experiments used can only provide a first guess. A discussion in the community would be helpful; perhaps we have better-equipped experiments for further studies. For the validation I propose eddy-covariance measurements with high resolution in time on the basis of a wavelet analysis (Schaller et al. 2019).

[Figure]

The authors highlighted problems like low level jets and breaking gravity waves that affect the turbulent exchange significantly, mainly after midnight. Closed chambers cannot usually measure (or only partly measure) these higher exchange rates. In the discussion I am missing another effect: at night, closed chambers have a longwave net radiation near zero. That means they always have neutral stratification, whereas outside a strong stable stratification exists (Riederer et al. 2014). Please discuss this effect too. In Figure 7b, before midnight such a radiation effect for the closed chamber may be possible. However, the virtual chamber cannot reproduce the radiation effect and I am probably seeing the influence of a strong stable stratification that is very typical before midnight, while after midnight it is often the case that condensation (dew) reduces the degree of stability. Some minor remarks:

Line 33: Recently eddy-covariance is used instead of eddy-correlation.

Line 109: I think sigma w is an important parameter to describe the influence of turbulence on fluxes. Because the basic instrumentation of the three experiments is a Bowen-ratio installation, please make some remarks about the measurements of sigma w.

Line 144: The linear gradient should be the first guess.

Line 145: Please explain the square root in connection with Figure 1 more clearly.

Fig. 4 and 7: Because I am colour-blind, I see nothing (only with scientific background and context am I able to form an impression). Could you please use black and grey instead of red and green?

The paper should be accepted with minor revisions.

References:

Riederer M, Serafimovich A and Foken T (2014) Eddy covariance – chamber flux differences and its dependence on atmospheric conditions. Atmos Meas Techn. 7:1057–1064.

Schaller C, Kittler F, Foken T and Göckede M (2019) Characterisation of short-term extreme methane fluxes related to non-turbulent mixing above an Arctic permafrost ecosystem. Atmos Chem Phys. 19:4041-4059.
* * *

---

## Author Comment (AC1) · 17 Apr 2020

We thank the Reviewer for his/her valuable comments. Please see below our answers to the reviewers. Various ways of improvement of the manuscript will be suggested based on these answers in order to adjust the manuscript in the way it was envisioned.

Response to the Reviewer #1:

The authors propose a novel approach to estimate fluxes exchanged between the surface and the atmosphere in nocturnal conditions. The method is based on a so-called Â′n virtual chamber approach ÂËŹz, which consists basically to estimate the storage term every five minutes and to infer its value in the absence of advection and turbulence. Under such conditions, it is supposed to equal the flux. As estimating

fluxes during nocturnal conditions remains a big challenge, there is a clear need for new approaches. The one proposed here has the advantage to not require additional measurements as those already provided by current flux towers. Only a recalculation of tracer concentrations and wind velocity means and variances on a five minute basis is required. It could thus be implemented at many different sites on existing data sets.

1. However, I see two big weaknesses in the approach, which let me think that it is not yet mature and that a strong reappraisal of the method is needed. First, if I understand well, the method relies on the hypothesis that vertical gas concentration profiles are flat (in the closed chamber option, Eq. 7), or linear (in the vented chamber option, Eq. 8). This is not realistic, vertical concentration gradients are generally much higher close to the surface than upward (see, for instance, profiles in Aubinet et al., 2005). This suggests that the method cannot be applied as it is in forests (where the problem of night flux estimate is the most critical) and is also probably questionable over shorter vegetation. Replacing a single point concentration by a profile measurement could certainly improve the method (Nicolini et al., 2018).

It's important to recognize that estimating emission rates of trace gases at night and from areas of limited size is a big challenge. There is therefore no argument about the claim that the methodology proposed is immature. However, it's an important first step as it allows us to address the surface exhalation question in conditions of stable stratification (such as over land at night and often over inland water bodies in daytime). We do not expect that our proposals will be widely accepted without independent examination of them. We are trying to initiate such external examination of what we suggest. A lot of thought and field work will be needed to clarify the circumstances under which the approach might work. It is clear that measurements at many heights would allow the matter to be clarified, and would permit the basic approach to be applied without assumptions regarding the average gradient within the layer of accumulation of emissions. This matter will be addressed in the revised manuscript in order to clarify under

which conditions the approach could work best.

Indeed, we have added the following text to the manuscript:

It is thought that the methodologies presented here will be found most useful in extraction of meaningful soil flux estimates in continuing strongly stable conditions, such as are often encountered over land at night and over inland lakes in daytime.

The assumptions made about the gradient in the layer of accumulation are based on theoretical expectations on the one hand (a linear decrease of concentration with height) and the closed-chamber approach on the other. There is no necessity to assume that either assumption replicates reality for any single time period, but rather that ensemble averages would yield behaviors close to one of these, or be bounded between them. Once again, we look for additional input on this matter.

2. Secondly, the method is not validated. The authors present flux estimates based on three measurement campaigns but they recognize that their method clearly under estimates the fluxes in the third campaign, due to a too low data acquisition frequency, and that the values they propose in the two first campaigns are only orders of magnitude. I think that a better validation could be provided by comparing their estimates with eddy covariance flux estimates captured during turbulent nights at the same site.

We acknowledge that the three field experiments that are discussed are certainly not the best tests of the virtual chamber hypothesis. However, it must be recognized that verification of the virtual chamber results will be difficult. The three datasets considered so far yielded results within the range of available field data. These findings need to be considered as a starting point.

Eddy covariance (and other micrometeorological methods) are largely inappropriate as sources of relevant verification material since such methods are limited by fetch constraints. They require sensors to be operated at heights typically many meters

above the top of the crop in question, and at night the corresponding footprint distance will be many hundred times this effective height. Relating eddy covariance results obtained above a given surface to that surface necessarily invokes the familiar criteria regarding fetch and time stationarity, neither of which is a highly relevant concern in the present virtual chamber context.

3. In addition, I'm surprised by the numbers they propose: at the Ohio site, their approach provides a flux estimate of 1-2 $\mu$gm-2s-1 which is 15-30 times lower than the average respiration rates observed at Fluxnet sites (Baldocchi et al., 2018). This is not totally impossible but appears in contradiction with another publication by the same team: O'Dell et al (2015) indeed reported that, at the same site, one plot emitted 146 g $CO_2$ m-2 on 104 days, which would correspond to an average of 16 $\mu$gm-2s-1. At the Zimbabwe site, the authors obtained an average of about 20 $\mu$gm-2s-1 , which is a rather realistic order of magnitude ; however, here again, the same team (O'Dell et al, 2018) reported at this site emissions of 197 and 235 g m-2 over 139 days, which corresponds to a three times higher average. It is clear that these values could not be compared directly as they are not taken at the same time scale. However, in view of the differences in the orders of magnitude, no indication is given that the virtual chamber approach provides reasonable flux estimates but I'm rather pushed to think that it underestimates the fluxes, which would be logical if vertical concentration gradients are not taken into account. A more detailed validation and, possibly, refinements of the method are thus needed, which seems possible in view of the available datasets from the Fluxnet or the ICOS networks.

The apparent incompatibility of the present CO2 results with estimates based on the application of conventional Bowen ratio energy balance (BREB) methodology (as by O'Dell et al., 2015 and 2018) is not surprising, since the latter assume that the measurements of gradients made above the surface of interest are indeed representative of the surface itself. A statistical method to avoid the problems that then arise (the Augmented Bowen Ratio Analysis, ABRA, q.v. Hicks et al., 2020) is now in development.

In due course, the results of the ABRA approach will yield a better basis for consideration of the present virtual chamber method. In the meantime, we are satisfied that the estimates presently derived are in line with expectations once allowance is made for the effects of soil temperature on CO2 emissions from soil. Note that the Zimbabwe dataset gives almost an order of magnitude higher CO2 effluxes, more in line with values reported from FLUXNET. We agree, however, that additional and independent verifications are needed.

It is our suspicion that centuries of farming of the Ohio soils might have depleted them of carbon to the point that effluxes are now much lower. The questions arising should be answerable when a complete annual cycle of relevant data becomes available.

The relevance of FLUXNET data is also questionable. We know of no dataset that includes measurements of CO2 concentration near the ground with 5-minute or 10-minute time resolution.

To clarify this matter, we have added the following text to the manuscript

In the lack of major upwind surface irregularities, these occurrences were attributed to the gravity wave phenomenon considered in detail by Blackadar (1957), but many alternative mechanisms appear possible (Mahrt et al., 2020). The key point of the Zimbabwe finding was that the site in question is at an altitude of more than 1400 m asl, and the occurrence of the CO2 build-up and nocturnal intermittency is a revealing indication of the ubiquity of the phenomenon.

We have also revised the following text in the manuscript

The methodology presented here diverges substantially from familiar micrometeorological strategies. First, it is focused on the ground itself (or the vegetation above it), and does not rely on the assumption that measurements made above the ground are indicative of the local surface. Second, the reliance on statistical methods to drive the analysis towards situations in which the prevailing stability is high but the wind speed is

zero reduces (if not eliminates) the conventional requirement regarding large fetches. Third, the method requires measurement with a time resolution such that intermittent bursts of turbulence can be identified and eliminated.

We have also added the following text to the manuscript

Hence, the small differences found in the studies are encouraging, although requiring additional research. A major outcome of the examination of available field data presented here is that the CO2 effluxes from the Ohio site (in November, as winter approached) were an order of magnitude less than the estimates derived in Zimbabwe (also in November, but approaching the warmest part of the year). Attributing this difference to the effect of soil temperature is appealing, but there are many other factors that remain to be explored.

4. I have also a remark concerning the statistical treatment: partial correlation coeffi-cients are computed and their time course is presented in the results but there is no mention on how they are taken into account in the study. Are they used as quality criteria? If yes, which thresholds are considered? Moreover, how do the authors justify the choice of these specific criteria (in place, for example of the standard error of ax and ay coefficients, which would provide a direct estimate on the flux uncertainty)?

We suspect that the reviewer is referring to the plots of Figures 4 and 7. These are not plots of partial correlation coefficients but of the quantity R1.232 derived from consideration of the various contributing variances and covariances. They are not used as a quality criterion. Moreover, it is the average daily cycles that are the final answer desired. We do not consider the time evolution in any other way than in showing the record of measured concentrations (Figures 3 and 6). The analysis does indeed yield estimates of the statistical errors on the fluxes computed. It was a conscious decision not to show these because the diagrams are already complicated.

The partial correlation coefficients are not used in any way other than as steps towards quantification of the most likely emission rate. The whole purpose of our statistical

treatment is to extract information from imperfect time series.

5. I have no specific comments on the paper structure and writing, which are both good. Just one remark concerning figure 1b: I suppose that the black line represents the CO2 concentration evolution with height and not its gradient as indicated on the line (the gradient is the slope of the line). This should be clarified.

We totally agree with the Referee that the black line in Figure 1b represents the concentration evolution of various gases (not only CO2) with height and not its gradient. Because in the open chamber approximation the depth of the affected layer increases with time but maintaining a constant gradient in the air. The figure 1b was corrected and clarified in the revised manuscript.

We revised the Figure 1 in order to explain the basis for this particular analysis better.

We have also added the following text in the manuscript

The model presented in Fig. 1 (a) is considered here to represent an extreme circumstance controlling the statistics that follow – a closed chamber. A second extreme is illustrated in Fig. 1 (b), intended to represent an open chamber. In this case, the depth of the affected layer increases with time, according to the flux from the surface, but maintaining a constant gradient in the air. If the surface emission rate Fc is constant, then the total accumulation in the growing layer ($\delta$m) will increase as the product $\delta z \cdot \delta C$. Since the change in height $\delta z$ is proportional to $\delta C$ (a linear dependence of C on z is assumed) and the efflux rate Fc is assumed constant, the increase of the mass of the constituent C must be proportional to $(\delta C)2$. In the closed chamber approximation, the change $\delta$m is proportional to $\delta C$, because the volume being filled is constant.

Please also note the supplement to this comment:
https://www.biogeosciences-discuss.net/bg-2019-393/bg-2019-393-AC1-supplement.pdf

[Figure]

[Figure]

[Figure]

*Figure 1. Schematic illustrations of the concepts now being explored. The x-type chamber simulation is represented to the left, leading to an approximation that the efflux at the surface can be derived from measurements of the rate of change of concentration with time. An alternative y-type extreme is represented to the right, in which the depth of the layer of relevance is allowed to grow while maintaining the same concentration gradient. The height of measurement of $\sigma_w$ is indicated as $z_a$.*

**Fig. 1.**

---

## Author Comment (AC2) · 17 Apr 2020

We thank Dr. Thomas Foken for his valuable comments. Please see below our answers to the reviewers. Various ways of improvement of the manuscript will be suggested based on these answers in order to adjust the manuscript in the way it was envisioned.

Dr. Thomas Foken points out that the analysis might benefit from additional attention to the stability regime. In accordance with this, the analysis has been repeated, with the conclusion that the amended Zimbabwe results do indeed appear more consistent. While doing this, the Ohio data were also examined from the same perspective, with no major changes resulting. However, the opportunity has been taken to modify both Figures 4 and 7 so that they do indeed represent the different datasets in the same

way. Previously Fig. 4(b) plotted a subset of the data for which confidence was highest, whereas Fig. 7(b) plotted all results. Now, all results are plotted in both diagrams. The results with highest confidence are summarized in the text.

Response to the Reviewer #2:

The authors present an interesting approach of using a "virtual chamber" to discuss the influence of the complicated condition of the atmospheric turbulence at night on chamber measurements. This is an important contribution to making chamber measurements more representative, even if some problems are still open like the validation. Furthermore, the design of experiments for virtual chambers must be updated. The three experiments used can only provide a first guess. A discussion in the community would be helpful; perhaps we have better-equipped experiments for further studies. For the validation I propose eddy-covariance measurements with high resolution in time on the basis of a wavelet analysis (Schaller et al. 2019). The nature of the problem is such that we would be most happy with an independent test. However, we are pursuing the matter ourselves, without yet discovering new datasets. It is our interpretation that the complexity of the circumstances in which our virtual chamber approach would work best is such that most mainstream meteorologists would take all possible steps to avoid it.

The authors highlighted problems like low level jets and breaking gravity waves that affect the turbulent exchange significantly, mainly after midnight. Closed chambers cannot usually measure (or only partly measure) these higher exchange rates. In the discussion I am missing another effect: at night, closed chambers have a longwave net radiation near zero. That means they always have neutral stratification, whereas outside a strong stable stratification exists (Riederer et al. 2014). Please discuss this effect too. In Figure 7b, before midnight such a radiation effect for the closed chamber may be possible. However, the virtual chamber cannot reproduce the radiation effect and I am probably seeing the influence of a strong stable stratification that is very typical before midnight, while after midnight it is often the case that condensation (dew)

reduces the degree of stability.

Certainly, the mixing regime inside a chamber will differ greatly from the outside air. This will be the case whether or not the air in the chamber is mechanically stirred. We want to avoid discussing this in detail, but the identification of the issue and its relevance to Figure 7(b) caused us to examine the data sets and their analysis again. The new scrutiny revealed instances of not-credible measures, constituting unexplained extremes in the data sequences. Hence, we chose to impose a data exclusion criterion. This entailed computing the averages and standard deviation of sequential packets of observations. If any particular data point differed from its preceding average by more than four standard deviations, it has now been excluded. This criterion has been applied to all of the sequences of data used. It had no consequences on the Ohio case, but resulted in the elimination of several data outliers in the Zimbabwe case.

The revised Fig 7 reflects the application of this criterion

We have also updated the following text in the manuscript

As before, there is no convincing reason to prefer the y-type results over the x-type, even though the near-zero results (x-type) indicated in the diagram are disturbing. If x-type and y-type results are averaged (as was the case in consideration of the Ohio dataset), the resulting estimate of the $CO_2$ efflux for the Zimbabwe November data is $20.7 \pm 4.8$ $\mu$g m-2 s-1 for the area sown with maize, and $24.3 \pm 2.5$ $\mu$g m-2 s-1 for the fallow. At the time of these measurements, the maize had not yet fully emerged and the fallow field was poorly vegetated (with weeds). Concentration and virtual temperature data refer (as before) to a height of about 1 m above ground level.

Some minor remarks: 1. Line 33: Recently eddy-covariance is used instead of eddy-correlation.

We have replaced the term "eddy-correlation" by the term "eddy-covariance" in the whole manuscript. 2. Line 109: I think sigma w is an important parameter to describe

the influence of turbulence on fluxes. Because the basic instrumentation of the three experiments is a Bowen-ratio installation, please make some remarks about the measurements of sigmaw.

Indeed, we have added the following text to the manuscript:

In practice, the wind speed u is an output of the sonic anemometer, as is the standard deviation of the vertical wind component $\sigma$w. The rate of change of concentration, dC/dt, is conveniently computed from the initial time sequence of measurement as:

dC/dt = $(C_{n+1} - C_{n-1})/(t_{n+1} - t_{n-1})$ (4) where the measurement level of C ($z_c$) is such that the lack of turbulence indicated by the sonic anemometer will also be indicative of a lack turbulent exchange at the height of measurement of C.

We have also added the following text to the manuscript:

The discussion above relates to a situation in which $\sigma$w data are routinely available, synchronized with the C measurements. Such data are regularly provided by modern three-dimensional sonic anemometers, whose deployment is usually associated with the determination of fluxes directly, by covariance between concentrations and the vertical wind speed component. In the present case, the requirement that deployment must follow guidelines for accurate flux determination can be relaxed, because it is only the magnitude of $\sigma$w that is needed. In Fig. 1 (b) the measurements are assumed to be at a height ($z_a$) above that of C-measurement ($z_c$). It is assumed that as $\sigma$w trends to zero at height $z_a$, so it does at a lower height $z_a$, or h. In the lack of measurements of $\sigma$w, virtual temperature gradients or TKE could be used equivalently.

3. Line 144: The linear gradient should be the first guess.

We agree that wind-tunnel (and other) experience suggests that the linear gradient is the better approximation. 4. Line 145: Please explain the square root in connection with Figure 1 more clearly.

The matter has been explained more carefully in the description of Figure 1.

5. Fig. 4 and 7: Because I am color-blind, I see nothing (only with scientific background and context am I able to form an impression). Could you please use black and grey instead of red and green?

As previously explained, the analysis has been repeated and Figures 4 and 7 has been corrected and clarified in the revised manuscript by using black and grey curves with solid and open circles.

We have also updated the following text in the manuscript

Consideration of only the most robust results (those for which R1.232 > 0.25) yield an average soil efflux rate of 2.55 $\pm$ 0.31 $\mu$g m-2 s-1 for the previously tilled surface, and 2.25 $\pm$ 0.32 $\mu$g m-2 s-1 for the untilled. The most likely averages of $CO_2$ nocturnal emissions from Ohio agricultural soils are therefore indicated to be about 2.5 $\mu$g m-2 s-1 for the conditions of the current test (November, after harvest), regardless of whether the surface was previously tilled.

Please also note the supplement to this comment:
https://www.biogeosciences-discuss.net/bg-2019-393/bg-2019-393-AC2-supplement.pdf

Figure 4. (a) The average variation during the night of the proportion of variance in dC/dt ('closed chamber'; solid circles) and (dC/dt)² ('open chamber'; open circles) that can be accounted for by the present statistical treatment. Black symbols relate to a previously-tilled area, grey to an adjacent area not tilled. (b) the estimates of surface efflux rates derived from the same analysis.

**Fig. 1.**

Figure 7. As in Figure 4, with black points relating to a fallow field and grey to an adjoining area recently planted with maize (on 8 November, 2013). The analysis procedure is identical with that leading to Figure 4. The period represented here is the first two weeks of the month of November. Solid points indicate results obtained using the closed-chamber approximation described here, open points represent vented-chamber approximations.

**Fig. 2.**

---

## Author Comment (AC3) · 17 Apr 2020

**References**

Blackadar, A. K.: Boundary layer wind maxima and their significance for the growth of nocturnal inversions, *Bull. Amer. Meteorol. Soc.*, 38, 283-290, 1957.

Hicks, B. B., N. S. Eash, D. L. O'Dell and N. S. Oetting, 2020. Augmented Bowen ratio analysis – I: Site adequacy, fetch and heat storage. Agric. and Forest Meteorol., in review.

Mahrt, L., Pfister, L. Thomas, C. K.: Small-scale variability in the nocturnal boundary layer. Boundary-Layer Meteorol., 174, 81-98, 2020.

O'Dell, D. L., Sauer, T. J., Hicks, B. B., Thierfelder, C., Lambert, D. M., Logan, J. and Eash, N. S.: A short-term assessment of carbon dioxide fluxes under contrasting agricultural and soil management practices in Zimbabwe, *J. Agric. Sci.*, 7, 32-48, 2015.

O'Dell, D. L., Eash, N. S., Hicks, B. B., Oetting, J. N., Sauer, T. J., Lambert, D. M., Logan, J., Wright, W. C. and Zahn, J. A.: Reducing CO2 flux by decreasing tillage in Ohio: overcoming conjecture with data, *J. Agric. Sci.,* 7, 1-15, 2018.